# Facilitators and barriers to implementation of suicide prevention interventions: Scoping review

Alexandr Kasal[1,2] [iD], Roksana Táborská[1], Laura Juríková[1,3],
Alexander Grabenhofer-Eggerth[4], Michaela Pichler[4], Beate Gruber[4],
Hana Tomášková[1,3] and Thomas Niederkrotenthaler[5]

[1]Department of Public Mental Health, National Institute of Mental Health, Klecany, Czech Republic; [2]Faculty of Social Sciences, Charles University, Prague, Czech Republic; [3]Department of Psychology, Faculty of Arts, Charles University, Prague, Czech Republic; [4]Department of Psychosocial Health, Gesundheit Österreich GmbH, Wien, Austria and [5]Center for Public Health, Department of Social and Preventive Medicine, Wien, Austria

## Overview Review

**Keywords:**
suicide; self-harm; interventions; barriers; implementation; facilitators

**Author for correspondence:**
Alexandr Kasal,
Email: alexandr.kasal@nudz.cz

## Abstract

We know that suicide is preventable, yet hundreds of thousands of people still die due to suicide every year. Many interventions were proven to be effective, and dozens of others showed promising results. However, translating these interventions into new settings brings several challenges. One of the crucial obstacles to success is not anticipating possible barriers to implementation nor enhancing possible benefits of factors facilitating the implementation. While we witnessed great support for suicide prevention activities globally in the past years, implementation barriers and facilitating factors are yet to be comprehensively mapped to help implementation activities worldwide. This scoping review maps current knowledge on facilitators and barriers to the implementation of suicide prevention interventions while using the Consolidated Framework for Implementation Research (CFIR) for classification. We included 64 studies. Barriers and facilitators were most commonly identified in the outer setting CFIR domain, namely in the sub-domain of patient needs and resources, which refers to the way in which these needs and resources are reflected by the reviewed interventions. The second most saturated CFIR domain for facilitators was intervention characteristics, where relative advantage, adaptability and cost of intervention sub-domains were equally represented. These sub-domains refer mostly to how the intervention is perceived by key stakeholders, to what extent it can be tailored to the implementation context and how much it costs. While intervention characteristics domain was the second most common also for barriers, the complexity sub-domain referring to high perceived difficulty of implementation was the most frequently represented. With reference to the results, we recommend adapting interventions to the needs of the target groups. Furthermore, carefully selecting the intervention to suit the target context concerning their adaptability, costs and complexity is vital for a successful implementation. Further implications for practice and research are discussed.

## Impact statement

While effectivity of different approaches and interventions to suicide prevention is well known, evidence on factors influencing their implementation is lacking. To the best of our knowledge, this is the first review of barriers and facilitators to implementation of suicide prevention interventions. By mapping these factors, this review identifies missing ingredients, which might make the difference between implementation failure and success. With reference to the CFIR framework, results from the review suggest that both the most common facilitator and barrier are patient needs and resources, representing almost fifth of all facilitators and almost third of all barriers. Several implications for practice and research can be drawn upon these results with the potential to bolster mental health of society and prevent unnecessary deaths. In line with the frequency of factors involving patient needs, we argue for person-centred approach to suicide prevention and involving individuals with lived experience in all stages of implementation process. This approach might lead to better implementation and better outcomes of prevention efforts. Future studies on implementation of suicide prevention activities should reflect on barriers and facilitators to expand the knowledge on these factors. The present review includes comprehensive Supplementary Materials, where all barriers and facilitators are described with reference to the original study and are listed according to seven types of interventions a) identification or screening of risk groups and public health surveillance, b) education of gatekeepers, c) effective treatment of mental disorders and follow-up of suicide attempters, d) means restriction, e) public health awareness, f) responsible media reporting and g) mixed interventions. These materials can be used in practice as well as in research to inform suicide prevention activities on a global level.

## Introduction

Suicide is a serious global public health issue. Globally, more than 700,000 persons die by suicide every year and the vast majority of suicides occur in low-and-middle-income countries (WHO, 2021b; Ilic and Ilic, 2022). Since 1990, age-standardised mortality rates for suicide have reduced but remain an important contributor to mortality worldwide (Naghavi and Collab, 2019). In 2017, suicide accounted for 1.4% of all deaths, with a global age-standardised mortality rate of 10.0 per 100,000 (He et al., 2021). Worldwide, men have higher suicide mortality rates than women. This accounts for all age groups, except for those aged 15–19 (Naghavi and Collab, 2019). Suicide is the leading cause of death in high-income Asia Pacific and is among the 10 leading causes of death in eastern/central/western Europe, central Asia, Australasia, southern Latin America and high-income North America (Naghavi and Collab, 2019).

Despite the high rates of suicidal behaviour, suicide is preventable (Mann et al., 2005; Zalsman et al., 2016). It has become an emerging priority for health care and is reflected in several policy documents (e.g., Comprehensive Mental Health Action Plan [MHAP] for years 2013–2030 [WHO, 2013]; Sustainable Development Goals [SDGs] of United Nations [UNDP, 2015]). Both policy documents set forth ambitious goals for suicide prevention that are aimed at reducing the number of suicide deaths, lowering the suicide rate by a third by 2030 and reducing the suicide mortality rate by 10% by 2030 as stated in the MHAP and SDGs. In order to achieve these goals, both MHAP and SDGs aim to deliver an effective prevention programme to those at risk for suicidal behaviour, including those with non-heterosexual orientation, transgender persons, youth and other vulnerable groups of all ages.

In spite of existing evidence and ambitious goals that were set to implement effective preventive measures, there are still gaps in implementing suicide prevention interventions globally. Although national strategies have been formulated and adopted, governments often underestimate the importance of coordinating the implementation procedures. Successful implementation is challenging, and it requires a change in the way people think and organisations operate. Factors such as limited knowledge of stakeholders about preventive measures, poor leadership, insufficient collaboration, lack of resources, equipment or staffing and political or social support can therefore represent barriers to a successful implementation (WHO, 2014).

According to WHO, the biggest barriers that need to be considered in more depth include management and logistics (i.e., understanding the problem, actions and interventions), stakeholders (i.e., leadership and management, teamwork and collaboration, legislation and policies), financial resources (i.e., budget for implementing suicide prevention), human resources, stigma, data collection or multisectoral involvement (WHO, 2018). The barriers represent a threat to the success and sustainability of suicide prevention interventions. They need to be identified to be overcome or even prevented. On the other hand, there are also several factors that might facilitate the implementation. These include adequate communication with stakeholders, policymakers and actors in charge of the implementation procedures; clear definitions of objectives and goals; initiatives to increase awareness and initiatives to build a monitoring system or design a national strategy (WHO, 2014). Generally, identifying barriers and facilitators and reflecting on them in the process of implementation is a common and useful approach to ensure a successful implementation in the field of

mental health (Langley et al., 2010; Davis et al., 2021; Erondu and McGraw, 2021; Webb et al., 2021; Le et al., 2022).

When tailoring an implementation strategy for the above-mentioned factors, it is important to have an applicable theoretical framework. Consolidated Framework for Implementation Research (CFIR) provides a set of constructs used in implementation research in health services. It was formulated with the goal to promote implementation theory development and evaluation of what works where and also why across different settings (Damschroder et al., 2009). It is widely used across different contexts (Kirk et al., 2015), including mental health care (Higgins et al., 2020; Mutschler et al., 2022). CFIR is organised into five different domains which reflect several aspects/actors/factors that are likely to influence the implementation process: a) *intervention characteristics* referring to properties of the intervention, b) *outer setting* referring to factors outside of the organisation such as policy context and needs of the patients, c) *inner setting* referring to the context of implementing organisation/body, d) *characteristics of individuals* related to qualities of those implementing and/or providing the intervention or service and e) *process* referring to how is the implementation planned, executed and evaluated. CFIR is described in depth elsewhere (Damschroder et al., 2009).

Number of strategical policy documents set ambitious goals, and many publications provide a comprehensive approach towards suicide prevention (WHO, 2021a; Ilic and Ilic, 2022). Hence, it is reasonable to expect wider implementation of suicide prevention actions worldwide. Yet, to the best of our knowledge, barriers and facilitators to implementation of suicide prevention interventions have not been comprehensively mapped, which might hinder implementation of suicide prevention actions. In this scoping review, we aim to systematically identify the implementation barriers and facilitators in suicide prevention interventions from already published peer-reviewed literature using the CFIR model for classification while also reflecting on the six typical suicide prevention categories (HHS, 2012; WHO, 2012).

## Methods

We followed the Preferred Reporting Items for Systematic reviews and Meta-Analyses extension for Scoping Reviews (PRISMA-ScR) Checklist (Tricco et al., 2018).

### Search strategy

We applied a broad eight-block search strategy constructed in Web of Science and Medline on 2 February 2020 to identify relevant articles published since inception of the databases (see Table 1). The following rationale for the eight-block strategy was applied. The first block ensures that suicide is included. While the second block is formulated to specifically include action of implementation, the third block is formulated to include implemented activity (e.g., programme, intervention). Fourth and fifth blocks comprise terms for barriers and facilitators. Sixth block includes terms that are not negatively nor positively framed (such as the fourth and fifth block, respectively) but still may act as a barrier or facilitator and, thus, should be included. Seventh block ensures that either the barrier- or facilitator-related terms are included. Finally, the last block combines the seventh block along with the theme on suicide and realisation of preventive measures (e.g., *intervention implementation*). We conducted a brief sensitivity analysis to test specificity of our search strategy by a) combining the second and third blocks as

**Table 1.** Eight-block search strategy used for identification of relevant studies

| # | Search command |
|---|---|
| #1 | (suicid*) |
| #2 | (implement* OR dissemin* OR difuss* OR "scal* up" OR applicat*) |
| #3 | (prevent* OR promot* OR program* OR avoid* OR plan* OR approach* OR practice* OR interven* OR guidelin*) |
| #4 | (obstacl* OR barrier* OR difficult* OR disincentive* OR hindrance* OR obstruct* OR restrict* OR limit* OR drawback* OR complicat* OR hurdle* OR problem* OR struggl* OR advers* OR roadblock*) |
| #5 | (facilit* OR ease* OR enabl* OR help* OR aid*) |
| #6 | (factor* OR mechanism*) |
| #7 | (#4 OR #5 OR #6) |
| #8 | (#1 AND #2 AND #3 AND #7) |

it could be too limiting if they were applied separately and b) omitting the sixth block as it could potentially only identify epidemiological studies on risk factors.

### Inclusion and exclusion criteria

We included studies that were published in the English language. Studies for inclusion had to be published in a peer-reviewed journal. Studies were included only if they reflected on the implementation process and mentioned either barriers or facilitators of implementation of suicide prevention interventions. Suicide prevention did not have to be the sole and main focus of included studies. However, it still needed to be mentioned. With reference to the study design, only literature reviews were excluded. We did not focus exclusively on studies that described effective suicide prevention interventions as the aim of this review is to identify barriers and facilitators to implementation in its full scope.

### Consolidated Framework for Implementation Research

We used the CFIR framework throughout the analysis and result description. CFIR has five different domains which reflect several aspects/actors/factors that are likely to influence the implementation process. Each domain has several sub-domains (see Figure 1): a) *intervention characteristics* (i.e., adaptability, costs and complexity of the given intervention), b) *outer setting* (i.e., the extent to which the intervention reflects the needs and resources of patients; the extent to which the implementing organisation closely collaborates with other relevant organisations; and level of support from external policies), c) *inner setting* (i.e., level of establishment of the given organisation; level of the organisation's absorptive capacity; availability of resources at organisational level), d) *characteristics of individuals* (i.e., attitudes and beliefs of staff engaged in implementation processes towards the intervention) and e) *process* (i.e., the extent to which the implementation is planned and executed; the extent to which it is possible to engage crucial partners such as community champions or opinion leaders in the intervention). CFIR is described in depth elsewhere (Damschroder et al., 2009).

### Inter-coder agreement

Two coders (AK and RT) applied an iterative process to screen 10% of all references (*n* = 280) to ensure that inclusion and exclusion criteria were clear. Agreement was calculated by dividing the number of agreements (*n* = 266) by the total number of reviewed references. There was a 95% agreement between the coders. Discrepancies (*n* = 14) were discussed until a full consensus was reached.

From the pool of included studies (*n* = 64), both coders independently extracted data from 10% of the randomly selected papers (*n* = 6). The data were extracted on all 26 variables (CFIR sub-domains, see Figure 1) across the 5 CFIR domains. For both, the CFIR sub-domains and CFIR domains, the total number of

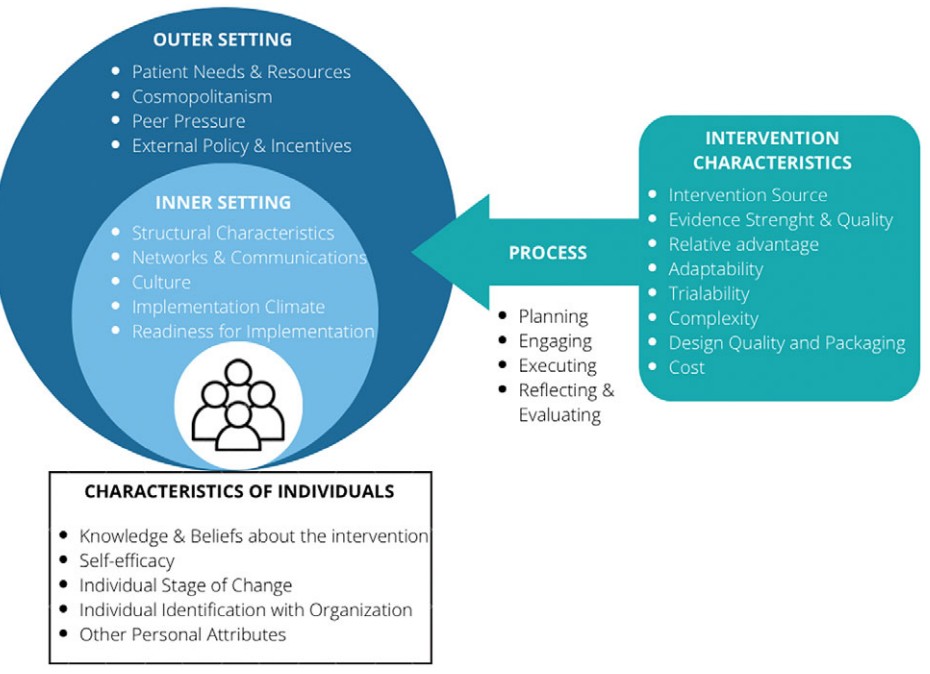

**Figure 1.** CFIR domains and subdomains. Figure adapted by The Center for Implementation based on original CFIR paper (Damschroder et al., 2009; TCI, 2022).

agreements was divided by the total number of identified factors (either barrier or facilitator; $n = 72$). There was a 93% agreement ($n = 67$) on domains and 85% agreement ($n = 61$) on sub-domains. This suggests that the level of agreement was decreasing the more the coders went in-depth of the CFIR framework. Regular meetings were introduced, where both coders discussed uncertainties, as these were the most common case of disagreement leading to a rather low inter-coder agreement for CFIR sub-domains data extraction.

### Data extraction and analysis process

For data extraction, we used a table where columns were denoted by the CFIR domains and rows were denoted by individual included studies. When extracting the data, we distinguished between facilitators and barriers to implementation. Every facilitator and barrier was assigned to a CFIR domain and CFIR sub-domain.

We computed descriptive statistics, both for the identified facilitators and barriers. These were expressed as counts ($n$) attributable to different CFIR domains and share (%) for CFIR domains for both facilitators and barriers. We also computed the same descriptive statistics for regional coverage by World-bank regions as well as by World-bank income typology of the countries (WB, 2022). We included this typology to assess the possibility of generalising our results across different contexts.

We used suicide prevention interventions typology suggested by WHO and HHS to structure the results (HHS, 2012; WHO, 2012). It includes six types of suicide prevention interventions: a) identification or screening of risk groups and public health surveillance, b) education of gatekeepers, c) effective treatment of mental disorders and follow-up of suicide attempters, d) means restriction, e) public health awareness and f) responsible media reporting. We

also introduced new category, where we included all studies that describe a mixture of the above-mentioned types.

### Protocol and methodological appraisal of included studies

This review was not pre-registered; protocol of the review was published in Figshare (Kasal, 2019). We did not appraise included studies for methodological quality as there was a plethora of designs spanning from randomised controlled trials to interpretative and case studies. Moreover, scoping review design usually does not include methodological quality as inclusion criteria (Arksey and O'Malley, 2005; Levac et al., 2010). However, we appraised included studies based on a hierarchy of evidence suggested for assessing feasibility (Evans, 2003), which is relevant to the implementation focus of this review. This hierarchy of evidence suggests a four-level typology: excellent, good, fair and poor (Evans, 2003).

### Results

We found 2,800 unique references. After two rounds of screening, we included a total of 64 articles (see FLOW diagram; Figure 2). Table with a brief description of all included studies can be found in Supplementary Materials.

### Sensitivity analysis

Combining second and third blocks resulted in 25,529 references. These were only extracted from Web of Science, suggesting that it is reasonable to have the blocks combined to achieve greater specificity of results. Omitting sixth block resulted in 2,103 references. Epidemiological studies and studies referring to risk factors were not a frequent reason for exclusion (see Figure 2). Hence, it is

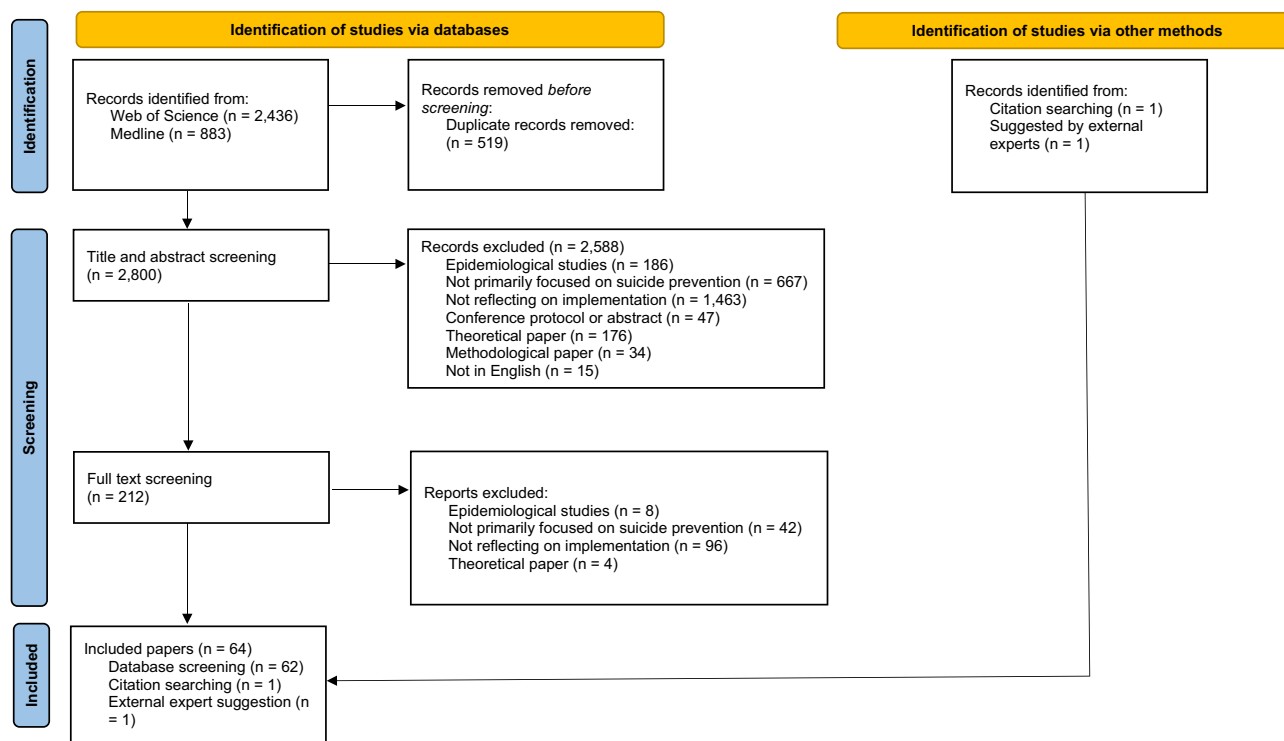

**Figure 2.** FLOW diagram adapted from Page et al. (2021).

reasonable to assume that this block has its specific function in search strategy.

## Descriptive statistics

We identified a total of 417 facilitators and 250 barriers in 64 different studies on implementation of suicide prevention interventions (Table 2). Studies were unevenly distributed over typology of interventions (see Supplementary Materials), and the same applies for facilitators and domains.

When taking CFIR into account, facilitators were most commonly found in *intervention characteristics* CFIR domain ($n = 152$; 36.5%), followed by *outer setting* ($n = 110$; 26.4%), *process* ($n = 68$; 16.3%), *characteristics of individuals* ($n = 45$; 10.8%) and *inner setting* ($n = 42$; 10.1%) domains.

With reference to barriers, *outer setting* ($n = 81$; 32.4%) was the most common domain, followed by *inner setting* ($n = 61$; 24.4%), *intervention characteristics* ($n = 52$; 20.8%), *characteristics of individuals* ($n = 41$; 16.4%) and *process* ($n = 15$; 6%) domains.

When considering the global coverage of included studies with respect to income, vast majority of studies were conducted in high-income countries ($n = 61$; 95%) and only a few were conducted in upper-middle-income countries ($n = 3$; 5%). No study originated from a lower-middle-income country or a low-income country.

World geographical regions were represented more evenly, although not all of them were represented. Most of the studies came from North America ($n = 28$; 43.5%), followed by Europe and Central Asia ($n = 21$; 33%), East Asia and Pacific ($n = 14$; 22%) and Latin America and Caribbean ($n = 1$; 1.5%) while three regions were not represented by any study: Middle East and North Africa, South Asia and Sub-Saharan Africa.

Half of the studies provided a fair quality of evidence ($n = 32$; 50.0%). Less than a third provided good quality of evidence ($n = 19$; 29.7%) and fifth of the studies provided poor quality of evidence ($n = 13$; 20.3%). None of the studies provided excellent quality of evidence according to the used typology (Evans, 2003).

## Narrative summary

The narrative summary of all 64 included studies structured by a) the type of suicide prevention intervention and b) facilitators/barriers that were rated as most prevalent is represented below. Full extraction table with all facilitators and barriers can be found in Supplementary Materials.

### Identification or screening of risk groups and public health surveillance

Included studies ($n = 8$) described two school-based screening programmes (Brown and Grumet, 2009; Evans et al., 2019), three screening programmes delivered in healthcare settings (Duffy et al., 2008; Lang et al., 2009; Snowden et al., 2019), two other programmes focused on youth: juvenile correctional facility-based screening programme and programme for native youth (Jacono and Jacono, 2008; Desmarais et al., 2012) and one study focused on delivery of internet-based mental health interventions (Batterham and Calear, 2017).

In total, we found 30 facilitators and 24 barriers to implementation (distribution across CFIR domains and sub-domains is described in Table 2). All studies were from high-income countries. Most of the studies were from the North American region ($n = 5$) followed by Europe and Central Asia ($n = 2$) and East Asia and Pacific region ($n = 1$).

*Facilitators.* Most facilitating factors were found in the *Intervention Characteristics* CFIR domain. Design quality and how the intervention was assembled was perceived as a very important factor. Namely, one study investigating preferences for internet-based mental health interventions found that providing screening, feedback on mental health symptoms as well as strategies on how to cope with them is appealing to users of these online tools (Batterham and Calear, 2017), potentially leading to higher uptake. Other frequent facilitator was knowledge and beliefs about the intervention in *Characteristics of Individuals* CFIR domain, referring to school–staff understanding the importance of suicide and self-harm prevention (Evans et al., 2019), subsequently easing the implementation of the screening. Similar facilitator was found in staff in the outpatient psychiatry setting (Lang et al., 2009).

*Barriers.* In included studies, barriers were most frequently identified in the *inner settings* domain. Specifically, availability of resources was the most common. In one school-based study, lack of human resources and lack of time available to spend time on the suicide prevention rather than actual teaching were mentioned as a barrier to implementation (Brown and Grumet, 2009). Additionally, lack of training was mentioned in other studies from school settings (Evans et al., 2019). *Outer settings* was another common domain in which barriers were found. They were related to strong preferences for tailored online mental health programmes in adult population, even if such tailoring would require extra time for assessment (Batterham and Calear, 2017).

### Education of gatekeepers

In this domain, we found 13 articles. Four of the included studies focused on programmes implemented in schools (Stein et al., 2010; Johnson and Parsons, 2012; Breux and Boccio, 2019; Chaniang et al., 2019). Gatekeeper training in community setting or among general public was the most common topic in four other studies (Capp et al., 2001; Evans and Price, 2013; Gask et al., 2019; Zeligman et al., 2019). Two of the included studies examined the curricula for native tribes (Dudgeon et al., 2017; Cwik et al., 2019), two other studies focused on gatekeeping in healthcare setting (Colombet et al., 2003; Inga-Lill and Danuta, 2004) and one study was concerned with implementation of guidelines for social work (Callahan, 1996).

Overall, we identified 97 facilitators and 75 barriers to implementation in the included studies (distribution across CFIR domains and sub-domains is described in Table 2). Most of the studies originated from high-income countries ($n = 11$), while the rest originated from upper-middle-income countries ($n = 2$). East Asia and Pacific region are represented by seven studies, North America by five studies and Latin America and Caribbean region by only one study.

*Facilitators.* One most common facilitating factor was found in *process* CFIR domain. It was associated with engaging with relevant individuals, actors and/or stakeholders. When working with native tribe community, both studies on this matter suggest that it is beneficial to engage with tribe's elders or local partner organisation with strong indigenous affiliation to ensure a successful implementation (Dudgeon et al., 2017; Cwik et al., 2019). Additionally, engaging with formally appointed internal implementation leaders was also found to facilitate implementation (Gask et al., 2019). Other frequent facilitator was the relative advantage of implementing the programme compared to other intervention or *status quo* (*intervention characteristics* CFIR domain). In included studies, it

**Table 2.** Counts of different domains and sub-domains of CFIR in respective types of suicide prevention intervention

| | | Gatekeeper education/training | | Effective treatment of mental disorders, follow-up of suicide attempters | | Identification/ screening of risk groups, public health surveillance | | Means restriction | | Mixed | | Awareness | | Media | |
|---|---|---|---|---|---|---|---|---|---|---|---|---|---|---|---|
| | | Facilitators | Barriers | Facilitators | Barriers | Facilitators | Barriers | Facilitators | Barriers | Facilitators | Barriers | Facilitators | Barriers | Facilitators | Barriers |
| Intervention characteristics | Intervention source | 7 | 1 | 3 | 0 | 0 | 0 | 0 | 0 | 2 | 0 | 0 | 0 | 2 | 0 |
| | Evidence strength and quality | 3 | 0 | 1 | 0 | 1 | 2 | 0 | 0 | 5 | 1 | 0 | 0 | 0 | 0 |
| | Relative advantage | 9 | 0 | 18 | 1 | 3 | 0 | 4 | 0 | 13 | 0 | 3 | 0 | 0 | 0 |
| | Adaptability | 8 | 2 | 10 | 3 | 2 | 2 | 5 | 0 | 4 | 0 | 1 | 1 | 0 | 0 |
| | Trialability | 0 | 0 | 0 | 0 | 0 | 0 | 0 | 0 | 0 | 0 | 0 | 0 | 0 | 0 |
| | Complexity | 2 | 8 | 4 | 7 | 2 | 2 | 1 | 4 | 2 | 5 | 1 | 1 | 0 | 0 |
| | Design quality and packaging | 3 | 1 | 8 | 2 | 7 | 0 | 0 | 0 | 4 | 1 | 1 | 0 | 5 | 0 |
| | Cost | 1 | 3 | 3 | 1 | 0 | 0 | 0 | 2 | 4 | 2 | 0 | 0 | 0 | 0 |
| | *Sub-total* | 33 | 15 | 47 | 14 | 15 | 6 | 10 | 6 | 34 | 9 | 6 | 2 | 7 | 0 |
| Outer setting | Patient needs and resources | 6 | 15 | 37 | 34 | 6 | 5 | 4 | 6 | 11 | 5 | 4 | 4 | 3 | 4 |
| | Cosmopolitanism | 5 | 0 | 7 | 2 | 0 | 0 | 0 | 0 | 9 | 0 | 0 | 0 | 0 | 0 |
| | Peer pressure | 0 | 0 | 0 | 0 | 0 | 0 | 0 | 0 | 0 | 0 | 0 | 0 | 0 | 0 |
| | External policy and incentives | 4 | 2 | 2 | 1 | 1 | 1 | 4 | 0 | 7 | 1 | 0 | 1 | 0 | 0 |
| | *Sub-total* | 15 | 17 | 46 | 37 | 7 | 6 | 8 | 6 | 27 | 6 | 4 | 5 | 3 | 4 |
| Inner setting | Structural characteristics | 3 | 5 | 0 | 3 | 0 | 1 | 0 | 1 | 0 | 0 | 0 | 0 | 0 | 0 |
| | Networks and communications | 4 | 3 | 0 | 0 | 0 | 0 | 0 | 0 | 0 | 0 | 0 | 0 | 0 | 0 |
| | Culture | 3 | 2 | 0 | 0 | 0 | 0 | 1 | 0 | 0 | 1 | 0 | 0 | 0 | 0 |
| | Implementation climate | 6 | 11 | 5 | 1 | 1 | 0 | 3 | 0 | 5 | 3 | 0 | 0 | 0 | 0 |
| | Readiness for implementation | 4 | 9 | 1 | 4 | 0 | 6 | 1 | 2 | 5 | 8 | 0 | 1 | 0 | 0 |
| | *Sub-total* | 20 | 30 | 6 | 8 | 1 | 7 | 5 | 3 | 10 | 12 | 0 | 1 | 0 | 0 |
| Characteristics of individuals | Knowledge and Beliefs about the Intervention | 6 | 3 | 9 | 8 | 3 | 3 | 5 | 2 | 3 | 7 | 1 | 1 | 2 | 4 |
| | Self-efficacy | 6 | 2 | 2 | 5 | 0 | 0 | 0 | 1 | 1 | 0 | 0 | 0 | 0 | 0 |
| | Individual stage of change | 0 | 0 | 3 | 1 | 0 | 0 | 0 | 0 | 0 | 0 | 0 | 0 | 0 | 0 |
| | Individual identification with organisation | 0 | 0 | 0 | 0 | 0 | 0 | 0 | 0 | 0 | 0 | 0 | 0 | 0 | 0 |
| | Other personal attributes | 0 | 1 | 2 | 2 | 1 | 1 | 1 | 0 | 0 | 0 | 0 | 0 | 0 | 0 |
| | *Sub-total* | 12 | 6 | 16 | 16 | 4 | 4 | 6 | 3 | 4 | 7 | 1 | 1 | 2 | 4 |

(*Continued*)

**Table 2.** *(Continued)*

| Process | | Gatekeeper education/training | | Effective treatment of mental disorders, follow-up of suicide attempters | | Identification/ screening of risk groups, public health surveillance | | Means restriction | | Mixed | | Awareness | | Media | |
|---|---|---|---|---|---|---|---|---|---|---|---|---|---|---|---|
| | | Facilitators | Barriers | Facilitators | Barriers | Facilitators | Barriers | Facilitators | Barriers | Facilitators | Barriers | Facilitators | Barriers | Facilitators | Barriers |
| | Planning | 2 | 1 | 0 | 0 | 0 | 0 | 3 | 0 | 5 | 0 | 1 | 0 | 0 | 0 |
| | Engaging | 11 | 4 | 4 | 2 | 1 | 1 | 0 | 0 | 16 | 1 | 3 | 0 | 2 | 0 |
| | Executing | 0 | 1 | 2 | 1 | 0 | 0 | 0 | 0 | 3 | 1 | 0 | 0 | 0 | 0 |
| | Reflecting and evaluating | 4 | 1 | 5 | 2 | 2 | 0 | 0 | 0 | 2 | 0 | 0 | 0 | 2 | 0 |
| | *Sub-total* | 17 | 7 | 11 | 5 | 3 | 1 | 3 | 0 | 26 | 2 | 4 | 0 | 4 | 0 |
| | Total | 97 | 75 | 126 | 80 | 30 | 24 | 32 | 18 | 101 | 36 | 15 | 9 | 16 | 8 |

was expressed as appreciating the programme and knowledge improvement on suicide prevention (Inga-Lill and Danuta, 2004; Johnson and Parsons, 2012).

*Barriers.* Most frequently mentioned barrier to implementation referred to patients' needs and resources (*Outer Setting* CFIR domain). Specifically, a study from Mexico City mentioned a few of these barriers which could hinder the implementation success. These include poverty of clients from the community service together with fees (although very small) for using the service, lack of support from family and stigmatising attitudes towards using the service, especially in the male population (Zeligman et al., 2019). Other study from a school setting suggests that lack of professional training is perceived as an unmet need by both school staff and students (Breux and Boccio, 2019). Other common barriers (*inner setting* CFIR domain) involved weak implementation climate resulting mostly from staff shortages in school settings (Breux and Boccio, 2019), restrictive care standards, as perceived by social workers, and disputes among staff due to different views on reforming standards of care (Callahan, 1996).

### Effective treatment of mental disorders and follow-up of suicide attempters

In this type of suicide prevention intervention, we identified 21 articles in total. The most commonly covered topic was implementation of therapies for those vulnerable to suicide or those after a suicide attempt in seven studies (Eccleston and Sorbello, 2002; Ellis et al., 2012a,b; Bosanac et al., 2015; Menefee et al., 2016; Awenat et al., 2018; Nicholas et al., 2019; Michaud et al., 2021). Other common topic was a follow-up of those with history of suicide attempt in six studies (Davis et al., 2009; Ghio et al., 2011; Luxton et al., 2014; Brovelli et al., 2017; Normand et al., 2018; Riblet et al., 2019), followed by four papers on implementation of e-health interventions (Buckingham et al., 2015; Bush et al., 2015; Boudreaux et al., 2017; Hetrick et al., 2017), two on implementation of guidelines (Baker et al., 2001; Campbell et al., 2011) and one on the management of acute risk (Draper et al., 2015).

In total, we identified 126 facilitators and 80 barriers to implementation (distribution across CFIR domains and sub-domains is described in Table 2). All studies originated from high-income country. Most of them were conducted in region of North America (*n* = 10), followed by Europe and Central Asia (*n* = 7) and East Asia and Pacific region (*n* = 4).

*Facilitators.* The most common facilitator was tailoring the programme to patient's needs and reflecting on their resources (*outer setting* CFIR domain). One study from the UK suggests that suicide-focused therapy can be accepted better by the patient if the therapy is tailored to the patient's past experiences. The therapy should make the patient feel understood with regard to his experiences and suicidal crisis and it should help the patient develop greater optimism (Awenat et al., 2018). Other study suggests that when administering computer-assisted safety planning, the assistance and guidance are often needed by the patients (Boudreaux et al., 2017). Also, in one study describing follow-up of patients after suicide attempt, weekly phone calls were perceived as satisfying and it was found that it was easier to recruit those patients who had already established contact with the service (Brovelli et al., 2017).

The second most common facilitator was relative advantage of the intervention or programme when compared to other interventions or *status quo* (*intervention characteristics* CFIR domain). It

was related to feasibility of telephone contact and high acceptance of follow-up of those with recent suicide attempt leading to low drop out (Brovelli et al., 2017). Furthermore, providing a brief qualitative information as a context to quantitative answers was found to be useful by both service users and practitioners within web-based systems for risk and safety management (Buckingham et al., 2015). Other study suggests that more intensive therapeutic programme can lead to quick establishment of a group cohesion (Eccleston and Sorbello, 2002).

*Barriers.* One of the most frequent barriers to implementation was the same as in facilitators – patient needs and resources (*outer setting* CFIR domain). This means that factors acting as facilitators can be barriers when not taken into account. This applies to cases when the intervention is not tailored to patients´ past experiences. In such a case, it fails to help them develop greater optimism (Awenat et al., 2018). In computer-assisted interventions, low computer literacy might hinder the implementation (Boudreaux et al., 2017). It was also pointed out that electronic alternative to therapeutic tool called "hope-box" could not provide a complete sensory experience when compared to traditional physical "hope-box" (Bush et al., 2015).

The second most common barrier was knowledge and beliefs about the intervention (*characteristics of individuals* CFIR domain). Study describing implementation of guidelines for the management of patients with depression among general practitioners suggests that cognitive dissonance theory might act as a barrier in the sense that when a guideline recommendation is in conflict with personal belief, the recommendation might be rejected (Baker et al., 2001). Other study reported on the uncertainty of help-line staff about confidentiality and data sharing with third parties such as emergency services if someone's life is at imminent risk (Draper et al., 2015).

### Means restriction
In total, we included three studies dealing with means restriction. One study described implementation of intervention aimed at firearms safety (Wolk et al., 2018), other study described counselling on lethal means in general (Betz et al., 2018) and the last study investigated measures at bridges and buildings (Hemmer et al., 2017).

In total, we found 32 factors facilitating and 18 factors hindering implementation of means restriction interventions (distribution across CFIR domains and sub-domains is described in Table 2). All three studies were conducted in high-income countries, while two of them were originally from North America and one from Europe and Central Asia region.

*Facilitators.* Most common facilitator of implementation in means restriction studies was adaptability of intervention (*intervention characteristic* CFIR domain). Before the implementation, both clinicians and patients reported that they would welcome some degree of adaptation of intervention on firearm safety, for example, streamlining screening via pen and paper into waiting rooms or variability in where to pick up locks to secure the firearm safety (Wolk et al., 2018). Other study suggests that there is some degree of adaptability in securing sites from which people might jump or fall and that it does not matter whether safety net or a barrier is erected (Hemmer et al., 2017). Other important facilitator was focused on the extent to which the intervention was in line with external policies and incentives (*outer setting* CFIR domain). This was particularly accented in a study on firearm safety in which authors argued that when the content of intervention is in line with key system priorities and recommendations made by relevant stakeholders, it could ease the implementation process (Wolk et al., 2018).

*Barriers.* Most frequently mentioned barrier was again found in patient needs and resources (*outer setting* CFIR domain). Study from the USA on firearm safety suggests that firearm ownership screening might infringe on patient's second amendment rights and that intervention would be less acceptable by those who keep firearms for protection, especially in areas with prevalent gun violence (Wolk et al., 2018). Other common barrier was the complexity of the given intervention (*intervention characteristics* CFIR domain). This barrier occurred in implementation of physical measures on bridges and buildings, where complex constructional issues are common and might complicate the implementation (Hemmer et al., 2017).

### Public health awareness
We found two studies describing implementation of (public) awareness raising intervention. One of them was focused on youth (Wasserman et al., 2012), and the other targeted male population (Dixon et al., 2019).

In included studies, we identified 15 facilitating factors and 9 barriers to implementation of public health awareness interventions in suicide prevention field (distribution across CFIR domains and sub-domains is described in Table 2).

Both studies were from high-income countries and were conducted in Europe and Central Asia region.

*Facilitators.* The most common facilitating factor was found among patients' needs and resources (*outer setting* CFIR domain). Study focusing on improving mental health in male population via football culture suggests that readiness and willingness to accept help are a crucial prerequisite for implementation success (Dixon et al., 2019). Youth's needs are also important to consider. For example, knowing the preference for role-playing as an opportunity to openly speak about mental health issues (Wasserman et al., 2012). The other frequent facilitator was engaging with target group in a way, which is natural and convenient for the target group, that is, via You Tube, Facebook and other social media, which are used among UK football fans (Dixon et al., 2019).

*Barriers.* The most frequent barrier was found in the same area as in facilitators – patient needs and resources (*outer setting* CFIR domain). Conversely to facilitators, when the needs and resources are not taken into account, it might hinder implementation of the programme. This was shown in a study, where the programme was administered only after school in some locations, thus competing with other free-time activities of the pupils (Wasserman et al., 2012). Similarly, too theoretical or time-limited sessions were not accepted as good as other sessions based on role-playing (Wasserman et al., 2012).

### Responsible media reporting
We found a total of four studies describing implementation of interventions in collaboration with the media. Three of the studies were concerned with guidelines for responsible reporting (Collings and Kemp, 2010; Roškar et al., 2017; Markiewitz et al., 2020). One study was dealing with developing social media suicide prevention messages for youth (Robinson et al., 2017).

In included papers, we identified 16 facilitators and 8 barriers to implementation of responsible media reporting (distribution across CFIR domains and sub-domains is described in Table 2). Again, all studies were conducted in high-income countries. Two of them originate from Europe and Central Asia and two from East Asia and Pacific region.

*Facilitators.* The most common factor facilitating implementation was design quality and packaging (*intervention characteristics* CFIR domain). One study suggests that media reporting guidelines have to find a compromise between comprehensiveness and brevity needed for everyday usage of journalists. It is also advisable to rather speak about recommendations than guidelines as the latter term can be perceived as limiting to journalist's freedom (Markiewitz et al., 2020). Furthermore, co-developing social media messages with suicide prevention content with intended target group members might lead to further reach of the campaign (Robinson et al., 2017).

*Barriers.* Most frequent barriers were patients' needs and resources (*outer setting* CFIR domain). In one study, this barrier was linked to the economic mindset of the journalism, especially related to celebrity suicide, which is contradictory to the guidelines recommendations, representing a barrier to implementation of such a recommendation (Markiewitz et al., 2020). The same study also suggests that when journalists feel like their professional freedom is threatened, it will severely hinder the implementation of the guidelines (Markiewitz et al., 2020). Second most frequent barrier was knowledge and beliefs about the intervention (*characteristics of individuals* CFIR domain), which was related to the fact that some of the journalists believed that following guidelines might create suicide-related taboo and pose a threat to public's right for unbiased information (Collings and Kemp, 2010).

### Mixed interventions

This section covers studies, which described implementation of suicide prevention actions related to more than one of the suicide prevention types. We found 13 of such studies. We grouped them according to the target group. Four of them covered implementation of interventions targeting general public and individuals vulnerable to suicide (Ho et al., 2011; Harris et al., 2013; Harris et al., 2016; Kongsuk et al., 2017), three studies focused on youth population (Whitney et al., 2011; Moutier et al., 2012; Apsler et al., 2017), two studies targeted members of native tribes (LaFromboise and Howard-Pitney, 1994; Stacey et al., 2007), two studies focused on veterans (Mills et al., 2006; Monteith et al., 2020), one study targeted family members of those vulnerable to suicide (Owens and Charles, 2017) and one study targeted elderly population (Kim, 2013).

In included studies, we identified 101 factors facilitating implementation and 36 factors acting as barriers to implementation (distribution across CFIR domains and sub-domains is described in Table 2). Vast majority of studies were conducted in high-income countries ($n = 12$) compared to single study from an upper-middle-income country. Slightly, over half of the studies come from Europe and Central Asia region ($n = 7$), while the rest are from the North American region ($n = 6$).

*Facilitators.* Most commonly, we found facilitators related to engaging (*process* CFIR domain) either with target group of an intervention or with those who implemented it. In a mixed intervention concerned with suicide prevention among veterans, knowing that there might only be a limited window of opportunity to intervene resonated with fellow veterans. It leads to higher motivation for providing needed help (Monteith et al., 2020). The same study also mentioned the importance of engaging with local veterans and firearm enthusiasts in order to effectively address the legal issues of firearm ownership and their rights (Monteith et al., 2020). Similarly, study describing dissemination of leaflet for families and friends concerned with possible suicide risk in their loved ones suggests that engaging with local champions in mental health field can promote dissemination of the relevant materials (Owens and Charles, 2017). Other common factor facilitating implementation was relative advantage of the intervention (*intervention characteristics* CFIR domain) compared to other intervention or *status quo*. Study exploring synergistic interactions in complex suicide prevention interventions in four European countries suggests that such a complex intervention can act as a catalyst for other activities and projects with a focus on suicide prevention. The synergies occur when complex interventions consisting of several suicide prevention approaches are implemented (Harris et al., 2016).

*Barriers.* Most frequent barrier was readiness of the organisations and other relevant structures to implement such interventions (*inner setting* CFIR domain). In the school environment, this was closely related to limited availability of time, staff and other resources which could be used for implementation of different school-based prevention schemes (Whitney et al., 2011). Similar issues also occurred in the primary healthcare settings where healthcare providers have demanding schedules and limited time to participate in trainings (Monteith et al., 2020). Same situation was mentioned in a study concerned with suicide prevention among elderly clients in social services (Kim, 2013).

Other common barrier to implementation was knowledge and beliefs about the intervention among those implementing it (*characteristics of individuals* CFIR domain). While working with older clients, novice workers were hesitant and nervous about triggering negative reaction when asked about suicidal ideation (Kim, 2013). Additionally, some school principals were sceptical about interest of parents, teachers as well as students themselves in the suicide prevention programme (Whitney et al., 2011). Furthermore, some principals questioned the boundaries between what is appropriate to handle in school and what is responsibility of the family, which posed as an apparent barrier to implementation (Whitney et al., 2011).

### Discussion

This review provided an overview of facilitators and barriers to implementation in suicide prevention interventions. We included 64 studies, which were conducted mostly in high-income countries mostly from North America and Europe and Central Asia region. To our surprise, we found more facilitators ($n = 417$) than barriers ($n = 250$). They were unevenly distributed across the CFIR domains.

Our results are in line with a recent study that suggests that factors such as staff uncertainty about effective ways to address suicidal behaviour in patients influence implementation of suicide prevention interventions across different healthcare settings (Davis et al., 2021). This study also points out that electronic health records

can simultaneously act as a barrier and facilitator. On the one hand, it makes it easier to engage in screening for suicide risk but on the other, it might be challenging to navigate the electronic tool (WHO, 2014; Davis et al., 2021). This applies to many of the factors that we identified in our review, which act both ways. In particular, web or electronically assisted tools or interventions tend to cause complications for those with low computer literacy and the opposite for those skilled in using electronic devices. Similar pattern applies to the patient needs and resources of CFIR sub-domain. When these needs and resources are taken into account, it leads to smoother implementation. On the contrary, when ignoring them, it might cause problems.

Other study on barriers and enablers to the implementation of recovery-oriented services suggests that staff training, public misconceptions of mental illness and joint working with families are the most salient factors (Erondu and McGraw, 2021). These factors were also found to act as either barriers or facilitators in the present review, suggesting that some factors are valid over different settings.

CFIR framework proved to be very useful when structuring results as it covers all important aspects of implementation act. Our results suggest that there is not a single element of implementation process, which could be overlooked. All the CFIR domains were represented by at least a dozen of facilitators or barriers.

Results from our review point out the importance of tailoring both intervention itself as the intervention process to specific settings and target groups. Designing specific implementation strategies and fine-tuning interventions content can, thus, lead to better implementation outcomes, with possibly lower resources needed. Below, for every type of suicide prevention intervention, we describe the way in which the most common facilitators and barriers can be reflected in future implementation efforts.

### Identification or screening of risk groups and public health surveillance

With reference to this approach to suicide prevention, our results suggest that the way in which a particular intervention is designed and assembled plays a crucial role in facilitating the implementation. This might be especially relevant for interventions that include screening processes in online environment, where several strategies can be used to tailor the intervention to meet the needs of each individual based on his gender, age and possibly other characteristics.

On the other hand, low availability of resources in the given organisation can hinder implementation efforts. Before initiating any implementation efforts, it is thus necessary to carefully review availability of money, training and education capacities, staff capacity and time. If these resources are not readily available, it can cause problems during the implementation phase.

### Education of gatekeepers

Our results indicate that when implementing interventions based on the education of gatekeepers, actively engaging with different actors is important for successful implementation. These actors can be manifold – opinion leaders, local champions, external change agents and of course members of the target group and their closed ones (e.g., parents in school settings). Conversely, when the needs and resources of the target group are not well reflected in the intervention, it can lead to complications (e.g., need for anonymity and low cost of services or enough time dedicated to education of gatekeepers to secure their self-confidence in providing the service).

### Effective treatment of mental disorders and follow-up of suicide attempters

Our findings further suggest that interventions that aim to treat mental disorders and/or follow-up on those who attempted suicide need to address patients' needs and resources. It plays a crucial role in both facilitating and hindering the implementation processes. If these needs are met (e.g., desired frequency of follow-up calls; attending therapy that reflects specific experience of each individual and aims to increase the patient's level of optimism), it can support the implementation processes. The opposite scenario might result in an unsuccessful implementation. It is thus recommended to map the needs and resources of the target group before any implementation is initiated.

### Means restriction

Our results indicate that interventions based on restricting access to the means can benefit from adaptability of the intervention. This means that intervention should in the best-case scenario provide few guiding principles, which can be tailored to the needs of the target environment no matter if it tries to restrict access to certain places, availability of firearms or pesticides. On the other hand, not reflecting the needs and resources of the target group in the intervention can lead to implementation failure. This can be illustrated by, for example, limiting access to paracetamol (which is commonly used for intentional self-poisoning) and causing unintended shortage of the drug availability for general population or by feelings of injustice by firearm owners when their availability or handling is limited.

### Public health awareness

Our findings suggest that when addressing public awareness through communication campaigns or other strategies, the needs and resources of the recipients must be addressed to ensure a successful implementation. This can be done by choosing appropriate communication channels (e.g., social media, printed materials, etc.) or appropriate settings for discussion (e.g., focus groups, role-playing). Conversely, if the intervention does not reflect the needs of specific target group, it can severely hinder the implementation (e.g., public awareness campaign aimed at senior population, which would take place exclusively on social media).

### Responsible media reporting

Suicide prevention interventions based on this approach often, at least partially, rely on guidelines for journalists. In order to strengthen suicide prevention efforts, journalists should follow the guidelines and engage in responsible reporting about suicide. Our results suggest that materials that are nicely designed and assembled to the specifics of journalist profession (e.g., need for brief materials with clear recommendations, suitable for fast pace of editorial work) can lead to smoother implementation. On the other hand, journalists are very sensitive to any kind of restriction on their professional freedom. Moreover, economic interests of media have an important role and might complicate the implementation of interventions that aim to deliver responsible reporting about suicide.

## Mixed interventions

For mixed interventions, our findings indicate that engaging with important stakeholders as well as target groups, champions and opinion leaders could have a positive influence on the implementation process. Conversely, low leadership engagement, low availability of resources and poor access to knowledge and information can hinder implementation of mixed interventions.

## Practical implications

Furthermore, our results might have a direct practical application. Different stakeholders and practitioners can take a detailed look on the specific type of intervention that they want to implement and can avoid barriers and/or reinforce facilitating factors. Specific citations extracted from included studies that were sought as either facilitators or barriers to implementation can be found in the Supplementary Materials.

## Call for person-centred suicide prevention

Single most common sub-domain identified in included studies was patient need and resources, acting as both a common barrier ($n = 73$; 29.2%) and a facilitator ($n = 71$; 17%). This CFIR sub-domain is rather vague, which was also the main criticism of the framework (Safaeinili et al., 2020). Our results suggest that splitting the factor into more sub-domains or making it a separate domain might be beneficial for further CFIR applications.

Nevertheless, high frequency of the patient needs and resources sub-domain should be a wake-up call for all public (mental) health professionals as well as policymakers and other stakeholders as it shows that the needs of the patient or client should be considered during all stages of implementation. Several authors already stressed the importance of person-centred suicide prevention approach (Duberstein and Wittink, 2015; Patel and Gonsalves, 2019). The present review provides new supportive arguments for this approach.

With respect to person-centred suicide prevention approach, individuals with lived experience should be involved in all phases of implementation. Despite the growing body of recommendations on involving people with lived experience in the implementation of suicide prevention (O'Connor and Portzky, 2018; SPRC, 2019), recent review suggests that involving people with lived experience in the implementation is still rare (Watling et al., 2022). The results from the present review and the frequency of unmet needs of patients further support the involvement of individuals with lived experience in the implementation processes.

## Limitations and strengths

The present review has several limitations. We searched only two of the main literature databases (Web of Science and Medline) which imply that our results might be limited. Furthermore, we did not search for grey literature which could result in a greater number of included studies. Similarly, holding discussions with experts from practice and individuals with lived experience could enhance our results by enabling us to collect qualitative information. However, this was beyond our capabilities. Although our results are mostly based on studies conducted in high-income countries and North America and Europe and Central Asia regions, they might still be applicable across other income and geographical regions. Furthermore, our review did not reflect the effectiveness of interventions described in the included studies. We are aware that there are many reviews on effectiveness of suicide prevention interventions, but none of them are focused on facilitators and barriers.

Strengths of our review are manifold: as far as we know, this is the first scoping review of facilitators and barriers to implementation of suicide prevention interventions. We applied broad search strategy which resulted in a rich pool of studies providing us with very interesting results. Based on their research design, included studies provided mostly fair or good quality of evidence and only one-fifth of the studies was rated as providing poor quality of evidence. In our view, the biggest strength is the possibility to translate our results from research into the public health practice as suggested above.

There are some implications for future research in the field of suicide prevention. Despite some limitations of the CFIR framework, we proved that the framework is applicable in the field and future studies should be aware of this framework. While effectivity is the most important factor when testing new or replicating established intervention, implementation facilitators and barriers also have a high value for practice. Therefore, we urge key stakeholders who are engaged in research on suicide prevention and practice to reflect on CFIR framework by at least one paragraph in future studies and reports, and mention the biggest obstacles they faced when implementing any intervention, and indicate factors that eased the process of implementation.

## Conclusion

Results from this review indicate diversity in barriers and facilitators to implementation of suicide prevention across six main types of interventions. For most of the interventions, patients' needs and resources were both the most relevant barriers and facilitators. Several implications for practice and research can be drawn upon these results with potential to bolster mental health of society and prevent unnecessary deaths. In line with the frequency of factors involving patient needs, we argue for person-centred approach to suicide prevention and involving individuals with lived experience in all stages of implementation process. This approach might lead to better implementation and prevention efforts might produce anticipated outcomes. CFIR was found to be a useful framework. We argue that the framework should be used in future implementation studies for the purpose of reporting facilitators and barriers. Despite its effectiveness, expanding the CFIR patient needs and resources sub-domain by adding more detailed information to the subject could improve its applicability.

**Open peer review.** To view the open peer review materials for this article, please visit http://doi.org/10.1017/gmh.2023.9.

**Supplementary material.** To view supplementary material for this article, please visit https://doi.org/10.1017/gmh.2023.9.

**Acknowledgements.** We would like to acknowledge kind help of Růžena Smrčková, Alena Znamenáčková and Kateřina Bílková in initial phases of the work on this review.

**Author contribution.** A.K. designed the whole study, reviewed the literature, extracted the data and wrote the first version of the manuscript. R.T. reviewed the literature and extracted the data. L.J. wrote introductory part and critically reviewed the manuscript. A.G.-E., M.P., B.G. and H.T. critically reviewed the manuscript and added to it. A.G.-E., M.P., B.G. and T.N. critically assessed and added to the data analysis and presentation of the results. T.N. supervised the whole process and provided several important comments, which significantly contributed to the overall quality of the paper.

**Financial support.** This study was supported by the project 'Sustainability for the National Institute of Mental Health', LO1611, Ministry of Education, Youth and Sports of the Czech Republic under the NPU I programme. The study was supported by the Charles University, Prague (A.K. grant number SVV 260596 and GA UK 552119). The funders had no role whatsoever in review design; in the collection, analysis and interpretation of data; in the writing of the report and in the decision to submit the paper for publication.

**Competing interest.** Alexandr Kasal, Roksana Táborská and Laura Bechyňová have been involved in preparation of Czech national suicide prevention policies and their implementation. Alexander Grabenhofer-Eggerth, Michaela Pichler and Beate Gruber are the team of the national coordination office for the Austrian suicide prevention programme SUPRA. Other authors declare no conflicts of interest.

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
