## [Reviewer Report]

Dear Editors,

I am pleased to submit our scoping review article entitled Facilitators and Barriers to Implementation of Suicide Prevention Interventions: Scoping Review. To our knowledge, this is the first comprehensive review of these factors.

In our paper, we are providing an overview of facilitators and barriers to implementing various suicide prevention interventions. In order to structure the results, we use six common types of interventions (e.g. means restriction, public health awareness etc.). We identified more than 63 implementation studies, which were conducted all over the globe, describing more than four hundred facilitators and more than two hundred barriers. Patient needs and resources were found to be both the most common barrier and facilitator. Among other things, our results suggest that tailoring preventive actions to the needs of the target group is the most crucial task for implementation success.

Further, the results of our review demonstrate that a plethora of factors might hinder or ease the implementation of suicide prevention interventions. We believe that the paper brings valuable information with several practical implications for researchers and public health practitioners all over the world.

Thank you for your time and for considering this research article.

Yours sincerely,

Alexandr Kasal

Mgr. Alexandr Kasal,

Department of Public Mental Health, 

National Institute of Mental Health, 

Topolová 738, 

Klecany, 

Czech Republic,

alexandr.kasal@nudz.cz

---

## [Reviewer Report]

*Comments to Author*: Dear Authors,

I had the opportunity to review your manuscript titled, “Implementation of suicide prevention: barriers and facilitators.” This topic is of great importance to the field of public mental health, and this manuscript holds promise to make an important contribution to our understanding of the barriers and facilitators to the implementation of suicide prevention strategies. As you argue in the manuscript, understanding the implementation science around suicide prevention is critical to ensuring that suicide prevention efforts succeed. I have some overarching comments and some specific feedback that I believe would strengthen your manuscript.

There are some problems with grammar throughout that made the manuscript difficult to read. I would recommend revising the manuscript with close attention to grammar. Additionally, throughout the manuscript, in the in-text citations, use a semicolon to separate different references.

A small comment about the abstract: The sentence, “The same CFIR domain of Intervention characteristics…” seems to be redundant with the previous sentence. If this is indeed the case, I would remove this sentence. If you meant to convey something new in this sentence, I would revise for clarity.

Introduction:

- I would recommend updating your WHO reference regarding the world-wide prevalence of suicide: https://www.who.int/publications/i/item/9789240026643

- Similarly, the reference for the LIVE LIFE guide should be updated to 2021: https://www.who.int/publications/i/item/9789240026629

- Throughout the manuscript, there are several instances of one-sentence paragraphs that can, in most cases, be combined with the previous paragraph.

- Capitalize CFIR when spelling it out (page 3).

Methods:

- It would be helpful to provide more details about findings/outcomes of the sensitivity analysis that you conducted.

- In the last sentence on page 4, it is unclear whether you included the studies that were not yet evidence-based in the review. If you did include these studies, I would conclude this sentence with, “…and so were included in this review.”

- Under the heading, “Inter-coder Agreement,” the sentence reading, “Agreement for these data was 85%...” is confusing. Was agreement 85% for CFIR subdomains and 93% for CFIR domains?

- On page 6, when you introduce the World Bank income typology, I would recommend explaining why you are including this. Are you aiming to approximate the resources available in a country that might be allocated to suicide prevention programming?

- I wondered if you differentiated between mental health symptom screening tools and those specifically designed for suicide prevention. This question arose for me in the second paragraph on page 6. I would add additional information to clarify your approach to determining whether articles should be included or excluded.

- You state that you did not appraise the quality of the studies you included in your review, as the main aim of the paper was to map the full scope of the topic. I do not think this is sufficient justification for not conducting a quality appraisal. I would report on the quality of each study in a table, but state that you are not excluding articles from the review based on quality assessment.

Results:

- Under the heading, “Facilitators,” Was the Batterham and Calear (2017) article targeting suicide prevention or mental health screening and coping skills training? Of course, there is a link between mental health screening, coping skills, and suicide prevention; however, related to my previous comment, you should make it more clear to the reader exactly what types of interventions you are including in your review.

- At the bottom of page 8, the implementation of “guidelines” seems to be different than the implementation of interventions. I think this goes back to your inclusion and exclusion criteria needing to be more clearly defined. This would help the reader to understand why you included the Callahan (1996) article.

I hope that these comments are useful to you should you be asked to revise your manuscript.

---

## [Reviewer Report]

*Comments to Author*: This paper is a scoping review of barriers and facilitators when implementing suicide prevention.

Abstract:

First part of the abstract gives a short and easily understandable perspective on the topic. Second part presents the results using the Consolidated Framework for Implementation Research (CFIR) as classification. I would suggest to try to “translate” more this part, in order to make it possible to understand the results also for readers unaware of the CFIR and its classification.

Introduction:

The introduction is easy to read and gives good information on the background. Some minor comments include:

• “First paragraph: this very recent reference could be added. Ilic M, Ilic I. Worldwide suicide mortality trends (2000-2019): A joinpoint regression analysis. World J Psychiatry. 2022 Aug 19;12(8):1044-1060. doi: 10.5498/wjp.v12.i8.1044. PMID: 36158305; PMCID: PMC9476842.)

• Last paragraph: “MHAP and SDG´s set ambitious goals and LIVE LIFE (WHO, 2021a, WHO, 2018a)”. This was stated at the beginning and could be adapted.

Methods

This section is well written and gives a good description of the study.

The Consolidated Framework for Implementation Research (CFIR) section repeats the a/b/c/d/e described in the Introduction. I would suggest (see also comment on the abstract) to avoid this repetition and rather illustrate these domains by an example which would make easier for the reader to understand them.

Regarding Protocol and methodological appraisal of included studies, I understand the idea to include all studies in order to cover the full range of the domain, but I think that their quality should be taken into consideration. While providing an individual appraisal of all studies at this stage is probably unrealistic, authors could shortly comment on their appreciation of studies methodological quality. This would help the readers to understand the results.

Results

No specific comment on the results/figures/tables.

Discussion

I think this section could be worked through and improved. Indeed, the Results section is quite long and not so easy to follow. Authors should therefore help the readers to identify the salient points of the results and comment on them. This could be done, for example, by commenting on the same subheadings as in the results (education/screening/gatekeepers etc…).

As I was involved in both studies, I noted that the paper Michaud L, Dorogi Y, Gilbert S, Bourquin C. Patient perspectives on an intervention after suicide attempt: The need for patient centred and individualized care. PLoS One. 2021 Feb 19;16(2) was not included in the review. It specifically studied the views on participants on what (do not) worked in an intervention included in the review (Brovelli et al, 2017) and could be considered.

In the discussion, I would also suggest to include the views of authors on how different stakeholders should be involved in the designing of the interventions, and especially on how patients could contribute to more adapted interventions, but also better public health actions or media interventions.

Conclusion

The conclusion section is very short; I think it could be expanded in order to synthesize key points identified by the authors and give a perspective to the reader, of how this study inform research and clinic.

---

## [Reviewer Report]

*Comments to Author*: Thank you for your submission of this manuscript on this important topic area. Overall, the peer reviewers found this manuscript to be strong; however, there were concerns regarding clarity and repetition in some areas. The Discussion and Conclusions, specifically, may benefit from deeper editing to make clear to the reader the most important findings.

---

## [Reviewer Report]

Dear Editors,

I am pleased to resubmit our scoping review article entitled Facilitators and Barriers to Implementation of Suicide Prevention Interventions: Scoping Review. To our knowledge, this is the first comprehensive review of these factors. We are grateful for the reviews, which helped to improve our review substantially. We substantially expanded the Discussion and Conclusion section. We also aimed to incorporate all of the received feedback on our paper.

In our paper, we are providing an overview of facilitators and barriers to implementing various suicide prevention interventions. In order to structure the results, we use six common types of interventions (e.g. means restriction, public health awareness etc.). We identified 64 implementation studies, which were conducted all over the globe, describing more than four hundred facilitators and more than two hundred barriers. Patient needs and resources were found to be both the most common barrier and facilitator. Among other things, our results suggest that tailoring preventive actions to the needs of the target group is the most crucial task for implementation success.

Further, the results of our review demonstrate that a plethora of factors might hinder or ease the implementation of suicide prevention interventions. We believe that the paper brings valuable information with several practical implications for researchers and public health practitioners all over the world.

Thank you for your time and for considering this research article.

Yours sincerely,

Alexandr Kasal

Department of Public Mental Health, 

National Institute of Mental Health, 

Topolová 738, 

Klecany, 

Czech Republic,

alexandr.kasal@nudz.cz

---

## [Reviewer Report]

*Comments to Author*: Dear Authors,

I have had the chance to review your revised manuscript and I believe that you did a nice job addressing my comments, and I hope that you found that they were helpful in strengthening your manuscript. The manuscript reads more clearly now. There remain a couple of grammatical issues/typos, but I believe these will be addressed at the proofs stage. I now recommend your manuscript for publication. Thank you for your important work on this critical topic.

---

## [Reviewer Report]

*Comments to Author*: Thank you for this revised version, which takes well into account the comments.

---

## [Reviewer Report]

*Comments to Author*: The Reviewers and I recommend the publication of this manuscript. However, several minor grammatical errors need to be attended to that won't change the paper's content. These are too numerous for me to list individually, but most fall into the category of missing articles. For example:

Page 3, last paragraph: Insert "the" before "WHO"

Page 4, second paragraph: Insert "the" before "Consolidated" 

Page 4, last paragraph: Insert "A" before "Number" 

...and so on.

I have been advised by the Editors that these errors can be fixed at the proofing stage.

Many thanks for this strong contribution to the literature!